# Forecasting severe grape downy mildew attacks using machine learning

**Mathilde Chen** [1] *, **François Brun**[2], **Marc Raynal**[3], **David Makowski**[4,5]

**1** Inserm U1153, CRESS, Epidemiology of Ageing and Neurodegenerative diseases, Université de Paris, Paris, France, **2** ACTA, INRA, UMR AGIR, Castanet Tolosan, France, **3** IFV, Bordeaux Nouvelle Aquitaine, UMT SEVEN, Villenave d'Ornon Cedex, France, **4** INRA, UMR Agronomie, AgroParisTech, Université Paris-Saclay, 78850 Thiverval Grignon, France, **5** CIRED, 94130 Nogent-sur-Marne, France

\* mathilde.chen@acta.asso.fr

**Data Availability Statement:** Data used in this study are summarized in Supporting information (see S1 Data).

**Funding:** This work received funding from the French Ministry of Agriculture (CAS DAR, SMART-PIC project), the Institut Carnot Plant2Pro (project

## Abstract

Grape downy mildew (GDM) is a major disease of grapevine that has an impact on both the yields of the vines and the quality of the harvested fruits. The disease is currently controlled by repetitive fungicide treatments throughout the season, especially in the Bordeaux vineyards where the average number of fungicide treatments against GDM was equal to 10.1 in 2013. Reducing the number of treatments is a major issue from both an environmental and a public health point of view. One solution would be to identify vineyards that are likely to be heavily attacked in spring and then apply fungicidal treatments only to these situations. In this perspective, we use here a dataset including 9 years of GDM observations to develop and compare several generalized linear models and machine learning algorithms predicting the probability of high incidence and severity in the Bordeaux region. The algorithms tested use the date of disease onset and/or average monthly temperatures and precipitation as input variables. The accuracy of the tested models and algorithms is assessed by year-by-year cross validation. LASSO, random forest and gradient boosting algorithms show better performance than generalized linear models. The date of onset of the disease has a greater influence on the accuracy of forecasts than weather inputs and, among weather inputs, precipitation has a greater influence than temperature. The best performing algorithm was selected to evaluate the impact of contrasted climate scenarios on GDM risk levels. Results show that risk of GDM at bunch closure decreases with reduced rainfall and increased temperatures in April-May. Our results also show that the use of fungicide treatment decision rules that take into account local characteristics would reduce the number of treatments against GDM in the Bordeaux vineyards compared to current practices by at least 50%.

## Introduction

Downy mildew is one of the most severe diseases of grapevines (*Vitis vinifera*). *Plasmopara viticola*, the pathogen responsible of this disease, is a heterothallic oomycete [1]. In autumn, winter eggs, called oospores, are produced. They overwinter in infected leaves, fallen to the vineyard ground [2]. In spring, they germinate as macrosporangium, which releases zoospores

L-i-cite, see https://www.instituts-carnot.eu/en/carnot-institute/plant2pro) and from the Bordeaux Vine Council (CIVB). This work is part of the #DigitAg project (ANR-16-CONV-0004, see https://www.hdigitag.fr/en/who-are-we/). The work of D.M was partly funded by the CLAND Institute of Convergence (16-CONV-0003, see https://cland.lsce.ipsl.fr/). The funders had no role in study design, data collection and analysis, decision to publish, or preparation of the manuscript

**Competing interests:** The authors have declared that no competing interests exist.

[2,3]. Zoospores are disseminated through rain splashes to young vines organs (leaves, flowers or young bunches), where they germinate and penetrate through stomata, causing primary infection after 7 to 10 days of incubation [3]. Sporangia, borne by sporangiophores, then emerge from affected host tissues. They are spread with wind and rain splashes to green parts of grapes, where they release new zoospores from asexual reproduction, which can then infect healthy tissues (secondary infection). *P. vitico*la damages on flowers and bunches lead to yield losses [2]. Leaf damage also induces a reduction in the sugar content, which induces a decline in the grapes quality [4].

For economic and health crop reasons, applying fungicide treatment remains a very common practice to control grape downy mildew (GDM) [5]. Several resistant varieties were developed [6], but they are still not used for the production of most of the more profitable wines, due to appellation regimes' specifications. Many microorganisms and botanicals were tested as an alternative to synthetic chemical fungicides [7–10], but most of them have not yet been developed for commercial purposes [5], mainly because of their low and unsteady efficacy in the vineyards.

Currently, many growers start spraying fungicides early in spring, and fungicide applications are then frequently repeated, about every two weeks in the Bordeaux region, a major vine producing area [11]. A large number of fungicide treatments are therefore applied over the course of the growing season, with implications for people living around vineyards health [12], grape growers' health [13–15], air [16], soil [17] and water [18] contamination, and entailing high production costs [19]. In 2013, an average of 18.5 pesticide sprays were applied in Bordeaux vineyards, 52% of which were used to control GDM [11].

Predictions of disease outbreak can assist farmers in decision-making for crop protection. Such predictions can be integrated into Decision Support Systems [20] or warning systems [21]. They could potentially be used in spring to estimate the incidence and severity of GDM at the end of the season. Based on model forecasts, growers could trigger fungicide applications only when the risk of GDM is high, avoiding unnecessary sprays. Models based on weather inputs can also be used to deal with more long term issues, for example to forecast GDM outbreaks under different climate conditions and assess the potential severity of the disease in the future.

In the past, several approaches were used to predict GDM epidemics. Historically, statistical models were first developed in Germany [22], France [23–25], Switzerland [26], Italy [27] and Australia [28]. Statistical models are simple to implement and they are able to predict complex systems, without explicating all functional mechanisms [29]. Mechanistic models differ from traditional statistical models by the need to translate every stage of the development cycle of an organism as functions; their structure makes explicit hypotheses about the biological mechanisms that drive infection dynamics [30]. This type of model relies on the estimation of many parameters and requires a good knowledge of the biological mechanisms and of the impact of different environmental variables on these mechanisms. Such models were also developed to dynamically predict primary infections of *P. viticola* [31]. Machine learning algorithms are also increasingly used in agriculture [32,33]. Machine learning models provide predictions of outcomes of complex mechanisms by relating outputs to inputs using very flexible algorithms [34]. On vine, Vercesi et al. [35] used such algorithms to estimate ability of *P. viticola* oospores to germinate. However, no study has been conducted to compare the performance of statistical and machine learning methods for predicting occurrence of high disease levels in vineyards, particularly for GDM.

Models performance depends on the model equations, on the accuracy of the parameter estimates, and on inputs used in the model. Since GDM development is influenced by several weather conditions such as rainfall and temperatures [36–39], climatic variables are frequently used in predictive models [22,31,39,40]. Other input variables can be included in predictive

models, such as crop cultivar or soil type. Field observations can also be used as input data for predictive models, e.g. in Savary et al. [41] and Delière et al. [42]. However, data collection can be time consuming and costly.

In this study, we assess the ability of statistical models and machine learning algorithms to predict the occurrence of high GDM levels at the end of the season, which has never been done before. More specifically, we develop different statistical and machine learning models to predict the risk of high GDM incidence or severity on leaves and bunches at the end of the season, in untreated Bordeaux vineyards. The models tested are generalized linear models [43], regularized regression models (LASSO) [44], and two machine learning algorithms, i.e. gradient boosting [45] or random forests [46]. These models are implemented with three sets of inputs, namely field scouting observations, climate inputs, and both types of inputs. Model performances are assessed by cross-validation using a large dataset of observations collected during 9 years in 153 vineyards in the Bordeaux regions. The most accurate models are then used to evaluate the potential reduction in the number of fungicide treatments achieved when model outputs are used to trigger fungicide applications. We also use our models to determine the impact of temperature increases and changes in precipitation on future GDM outbreaks.

## Material and methods

### Data

Grape downy mildew (GDM) incidence and severity data were collected in several vineyards located in the Bordeaux region by the French vine and wine extension service (*Institut Français de la Vigne et du Vin*, IFV). Data have been collected with the agreement of the winegrowers. Different vineyards were included in the dataset each year. A site-year is a unique combination of vineyard site and year. The total number of site-years included in the dataset was 153. Each monitored site-year consisted in an untreated row of vines including from 6 to 165 plants and further referred to as "plot". Each untreated row was surrounded by two other untreated rows, to ensure that they were not unintentionally sprayed with fungicide. In the monitored central row, weekly visual inspections were performed on grape stocks, leaves and bunches in order to assess disease incidence and severity. The proportion of vine stocks, leaves and bunches displaying symptoms (incidence) and the average percentages of leaves and bunches necrotic area (severity) were recorded. In total, 1 to 19 visual inspections were conducted in each vineyard. Observations were conducted from budburst (i.e. week 10, early March) until at least bunch closing (i.e. week 29, mid-late July) or stopped when the proportions of infected vine stocks and bunches were close to 100%. For each plot, the level of GDM (i) leaf incidence, (ii) bunch incidence, (iii) leaf severity and (iv) bunch severity at the end of the season were derived from the last epidemiologic observations. Among the plots, the median GDM incidence in the dataset was 16.7% and 21.4% on leaves and bunches, respectively. The median GDM severity reached 3.0% and 4.0% on leaves and bunches, respectively. Contrasted outbreaks were observed between 2010 and 2018 seasons. For example, the disease incidence on leaves ranged from 0 to 54.8% in 2011 and ranged from 2.2 to 100% in 2018 (Fig 1A).

Dates of GDM onset were estimated by analyzing incidence data on vine plants, i.e. the proportion of infected plants in a plot. GDM onset was defined as the first week in which the proportion of infected vines stocks exceeded 1%. The number of weeks between the first week of the year and this date was estimated for each plot by survival analysis in order to deal with censored data [47]. Censored GDM onset dates were found in 95 plots (right censored: 38.9%, left censored: 38.9% and interval censored: 22.1%). Censored data were imputed by a semi-

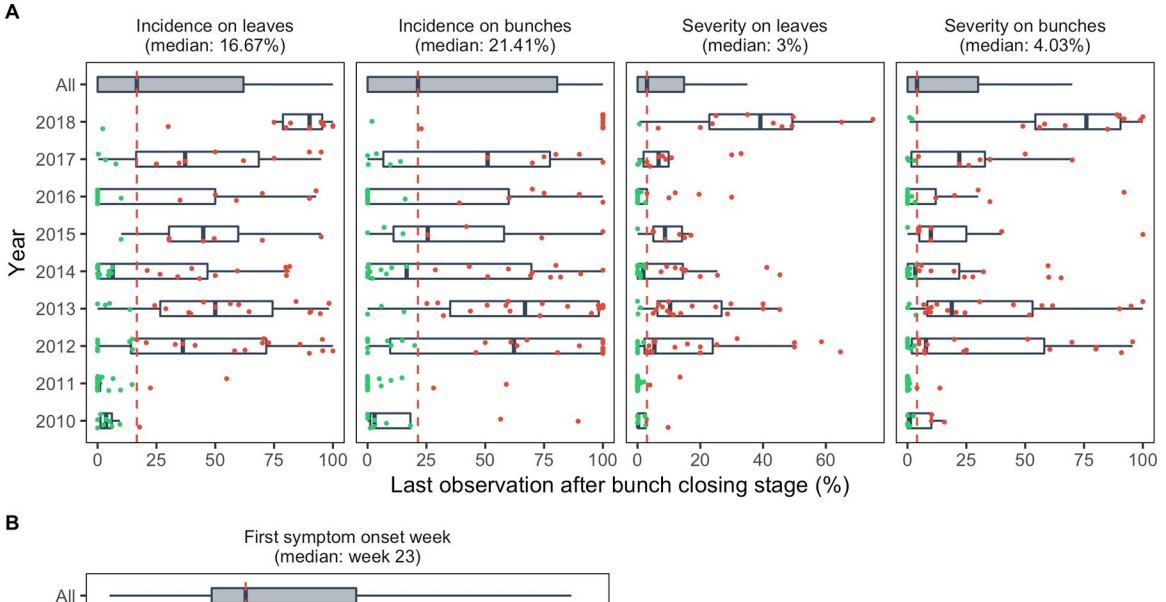

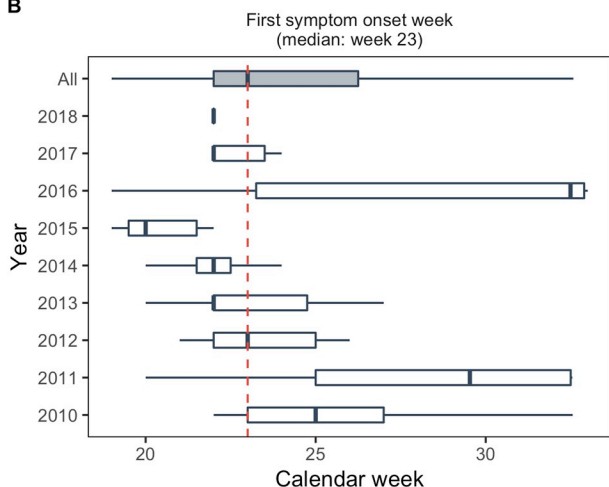

**Fig 1. Grape downy mildew (GDM) outbreaks temporal variability in monitored untreated plots.** (A) Last value of GDM incidence and severity on leaves and bunches after bunch closing in monitored plots. The color of the point represents the health status of each plot at the end of the season (green = last observation < median; red = last observation ≥ median). (B) Imputed disease onset dates in monitored plots. Median contamination levels and median disease onset date are represented by vertical dashed lines in panel (A) and panel (B), respectively. In both panel, the lower and upper hinges of the boxes correspond to the first and third quartiles (the 25th and 75th percentiles) and horizontal segment represent the range between min and max values.

parametric survival model [48] including the average rainfall between March and June as covariate [47]. In plots where few observations were collected, the imputed dates of disease onset were close to the median onset date of the dataset, which corresponded to week 23, i.e. early-mid June. Between 2010 and 2018, GDM onset date varied across years. For example, less than 25% on the monitored plots displayed symptoms by week 23 (i.e. early-mid June) in 2010, 2011 and 2016, whereas 100% of the plots were infected at this date in 2014, 2015 and 2018 (Fig 1B).

Climatic variables were computed from the SAFRAN database, produced by the French national meteorological service (*Météo-France*). SAFRAN data covers France in the form of an 8 by 8 km grid [49]. For each plot and each year, mean amount of rainfall (in mm day$^{-1}$) and mean temperature (in °C) were calculated in March, April, May and June from the weather data of the grid cell including the considered plot (Fig 2). Contrasted climatic conditions were observed between years. For example, the average precipitation measured in March 2012 and

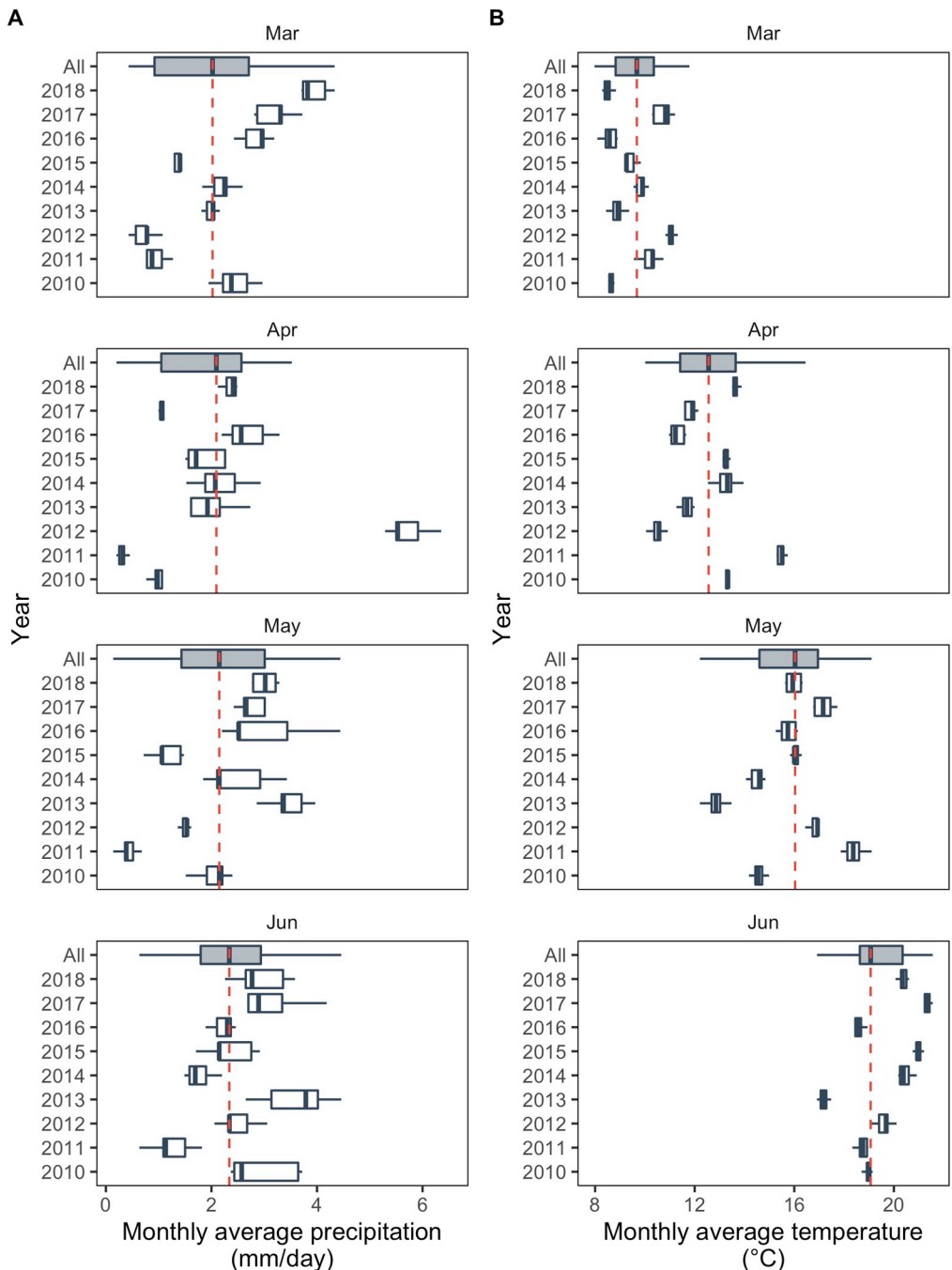

**Fig 2. Climatic variability in March, April, May and June during the 2010–2018 period in the 153 untreated monitored plots.** (A) Mean monthly precipitation amount (in mm day$^{-1}$) and (B) mean monthly temperature (in ˚C). Median precipitation amount and median temperature are represented by vertical dashed lines in panel (A) and (B), respectively. In both panel, the lower and upper hinges of the boxes correspond to the first and third quartiles (the 25th and 75th percentiles) and horizontal segment represent the range between min and max values.

in March 2018 was 0.7 mm day$^{-1}$ and 3.8 mm day$^{-1}$, respectively (Fig 2A). At these dates, the average temperature was 11.0˚C and 8.5˚C (Fig 2B). Data used in this study are summarized in Supporting information (see S1 Data).

## Models predicting occurrence of high levels of disease incidence and severity

Different models were considered to calculate the probability to reach a high level of contamination at bunch closing stage, i.e. higher than the median value reported in the dataset. Four types of output were considered in turn, i.e., incidence on leaves, incidence on bunches, severity on leaves and severity on bunches. For each output, four types of models were developed in the version 3.5.1 of the statistical software R [50] and compared to predict occurrence of high level of GDM (i.e., higher than median): generalized linear models (binomial-logit models), further denoted as GLM [43], binomial LASSO regression [44], random forest [46] and gradient boosting [45]. Depending on the considered output, the predictions returned by these models correspond to predicted probabilities of high levels of incidence or severity of GDM on leaves or on bunches.

Three binomial-logit models were developed: one model including a single input i.e., disease onset date, one model taking into account monthly average precipitation and temperature in March, April May and June, i.e. eight weather inputs, and one model including both disease onset date and weather variables as inputs, i.e. nine inputs variables. The models were fitted to data using the glm R function. The most relevant inputs of the last two models were selected using a stepwise procedure based on the Akaïke criterion (AIC) implemented with the R function stepAIC from the MASS R package, version 7.3 [51].

LASSO regression, random forest and gradient boosting were first fitted using weather inputs only, and then using both weather inputs and the GDM onset date. LASSO regression (implemented here with a logit link) is a special type of regression model fitted using a penalty term shrinking regression coefficients towards zero. Here, this model was fitted with the version 2.0 of the R package glmnet [52] and the most relevant model inputs were selected by cross-validation using the R function cv.gmlnet. Random forest and gradient boosting are ensemble learning algorithms; they are based on the combination of multiple learning simple algorithms to improve prediction performance. Random forest is a bagging method developed by Breiman [46]. The algorithm builds an ensemble of independent deep decision trees (500 in our study) from bootstrapped samples. Deep trees have the properties to have low bias but high variance, and when combined together, produce an output with lower variance. In the case of gradient boosting, an ensemble of successive shallow trees are built (100 in our study), such as each new tree predicts the residuals of the previous one. Random forest and gradient boosting were fitted using the R packages ranger, version 0.11.2 [53] and gbm, version 2.1.5 [54] to predict occurrence of high levels of GDM.

## Model assessment and sensitivity analysis

The ability of the fitted models to predict occurrence of high level of GDM was assessed by year-by-year cross validation using the area under the ROC curve as a measure of classification performance [55,56]. Here, we used a year-by-year cross-validation to account for the strong "year" effect on the disease intensity. As data collected during the same year in different plots are not independent, it was safer to remove all the data collected a given year at each cross-validation step. This is equivalent to a group-wise cross-validation based on 9 groups corresponding to the 9 years of data included in the dataset.

A separate ROC analysis was conducted for each output and each model separately. The 153 plots were divided into two subgroups depending on whether the final disease observation (Y) was above the regional median value (Yt), computed for the 2010–2018 period, or less than or equal to this threshold. The probability of high level of GDM, i.e. Y > Yt, was then estimated by each model for each plot in each subgroup. Let I denote the prediction of

a given model in a given plot, i.e., the predicted probability of high level of GDM in a given plot. Each value of I was compared with a decision threshold It. The results were used to determine the true positive proportion (TPP) (number of plots with I > It, in the subgroup defined by Y > Yt divided by the total number of plots in this subgroup) and the true negative proportion (TNP) (number of plots with I ≤ I t, in the subgroup defined by Y ≤ Yt divided by the total number of plots in this subgroup). TPP and TNP are estimates of P (I > It | Y > Yt) and P(I ≤ It | Y ≤ Yt) and are referred to as "sensitivity" and "specificity", respectively. The ROC curve of model is a graphical plot of sensitivity against 1-specificity. The values of TPP and TNP are calculated by allowing the decision threshold (It) to vary over the range of its possible values. A ROC curve that passes close to the point (0, 1) shows that the model gives satisfactory results in terms of sensitivity and specificity. A choice can thus be achieved by using the model with an appropriate choice of the decision threshold. A ROC curve that passes close to a straight line joining the points (0, 0) and (1, 1) shows that the model is non-informative (i.e. no better than a random decision). This approach is common in phytosanitary studies when the objective is to discriminate between low and high levels of infection [57–59]

A useful summary of the overall accuracy of a model is the area under the ROC curve (AUC). AUC is expected to be equal to 0.5 for a non-informative model, and to 1 for a perfect model. TPP and TNP values were used to compute the sensitivity, specificity, and AUC for each output variable and each model. The computations were performed using the pROC [60] package of the version 3.5.1 of R statistical software.

For machine learning algorithms, inputs were ranked according to their importance. For random forests, the importance corresponds to the increase in the misclassification frequency after random permutations of the values of each input. The variables leading to the largest average increase of misclassification frequency are considered most important [46]. For gradient boosting algorithms, the average improvement of the loss function (generally the MSE for regression or the deviance for classification) made by each variable is computed. The variables with the largest average improvement, i.e. the relative influence, are considered most important [45].

A sensitivity analysis was conducted to explore the potential consequences of an increase of temperature and/or of a change in rainfall. The best model among the models including weather input variables (i.e., the model with the highest AUC) was used to compute the probability of high levels of GDM according to different climatic scenarios. The original temperature data were increased by +1˚C, 2˚C, 3˚C or 4˚C, successively. Rainfall was increased by +5%, 10% or 15%, and then decreased by the same levels. All combinations of temperature and rainfall changes were considered. The selected model was run using each climate scenario.

A graphical summary of the modeling framework is presented in Fig 3.

## Number of fungicide treatments

The probability of severe attack was estimated for each plot as a function of the GDM onset date using the GLM model. We calculated the number of fungicide treatments against GDM resulting from the use of model output (i.e., predicted probability of high GDM) to trigger treatments. To do so, we assumed that fungicide treatments were applied only in plots where the probability of severe attack was higher than a predefined threshold and that vine growers apply fungicide every two weeks after the first treatment until late August (week 35) [61]. Several probability thresholds in the range 0–1 were considered successively, and the resulting number of treatments was calculated for each threshold and each plot. The average number of treatments over plots was finally calculated for each probability threshold. This number was compared to the mean number of GDM fungicide treatments applied in 2010 and 2013

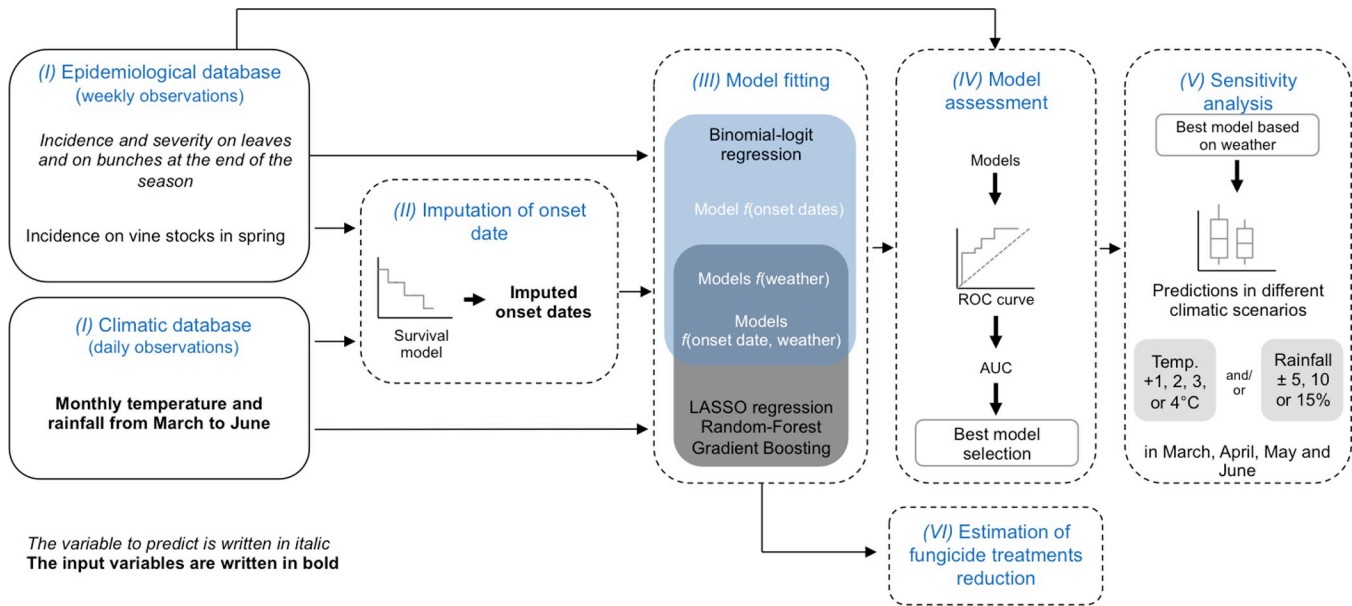

**Fig 3. Illustration of the modeling framework implemented in this study.** This modeling framework included 6 steps. (I) Extraction of epidemiological and climatic data from two different databases (model input and outputs are written in bold and in italic, respectively). (II) GDM onset date imputation by a semi-parametric survival model. (III) Models fitting. (IV) Models assessment based on a ROC analysis; models with higher area under the ROC curve (AUC) were selected. (V) Sensitivity analysis of the model outputs to weather inputs. (VI) Estimated reduction in GDM fungicide application obtained by delaying the date of the first fungicide application compared to current practices in the Bordeaux vineyards; calculations were based on GLM model predictions using dates of appearance of GDM as inputs.

according to the results of a survey conducted by the French Ministry of Agriculture's Statistics and Prospective Service (SSP) [11].

## Results

### Ability of the models to distinguish between high and low levels of disease

The best models according to AUC are those including the full set of inputs i.e., both climate variables and the date of disease onset (Fig 4). The highest AUC (0.86) is obtained with gradient boosting for incidence on leaves. For the three other outputs, i.e. leaves severity, bunches incidence and bunches severity, the AUC values of the best models were between 0.78 and 0.85.

Model performances are decreased when climate variables are omitted for predicting incidence and severity, but the AUC always remains very close to or slightly higher than 0.75 for all outputs. The omission of the date of disease onset from the set of inputs has a strong impact on the performance of the models. However, when this variable is omitted, the AUC of best models remains higher than 0.70 for incidence on bunches and is even higher than 0.75 for incidence on leaves (AUC = 0.77). The decrease of AUC resulting from the omission of the date of disease onset is stronger for severity on leaves and for severity on bunches for which the best AUC values are lower than 0.70 (0.67 and 0.65, respectively) (Fig 4).

Considering the models including climate inputs only (second row of Fig 4), gradient boosting shows the best performance for three of the four outputs (incidence on leaves, incidence on bunches, severity on bunches). The AUC values of random forest are, however, very close. The other types of models (LASSO, GLM with and without input selection) show contrasted results depending on the output. For example, LASSO has a high value of AUC for incidence on leaves but gets a very low value for severity on leaves. Very variable AUC values are also obtained for GLM.

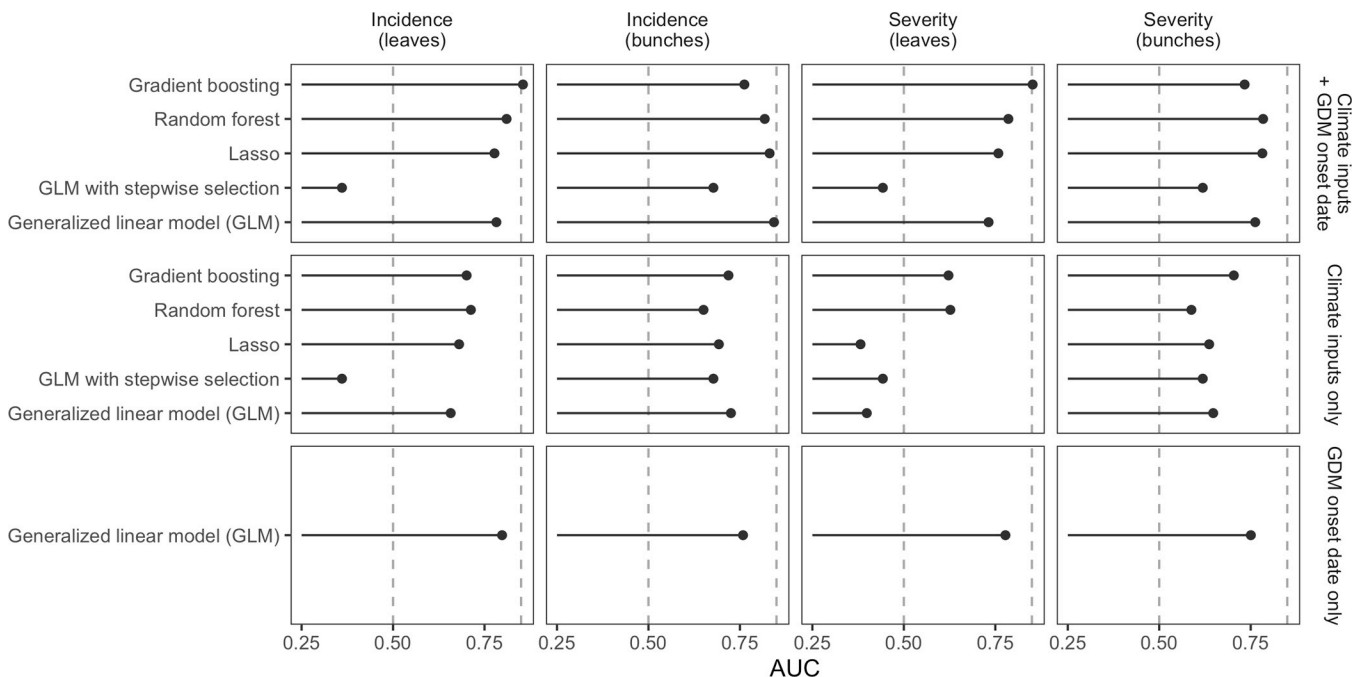

**Fig 4. Area under the ROC curve (AUC) of several models used to predict occurrence of high level of GDM incidence and severity on leaves and bunches.** Higher AUC represents higher model performance.

Considering the models including climate inputs plus date of disease onset (third row of Fig 4), gradient boosting has the highest AUC for two outputs, namely incidence and severity on leave. Random forest, LASSO and GLM without selection also show good performance. Stepwise input selection substantially decreases AUC values revealing that this selection procedure was unable to select relevant inputs. Better results are thus obtained with GLM without input selection.

## Importance of the model inputs

The inputs are ranked according to their importance in Fig 5 (see also S1, S2 and S3 Figs).

The date of GDM onset is ranked first for all outputs (Figs 5A and 5B and S1, S2 and S3). This input has thus a stronger impact on model classifications than all the considered climate inputs. The difference of importance between the date of disease onset and the most important climate input is stronger for random forest than for gradient boosting but the date of disease onset is ranked first with both approaches.

The most important climate inputs are those related to precipitation in late spring, i.e. mean precipitation in May (for random forest and gradient boosting) and in June (for gradient boosting). The least important variables are the average temperature in March for random forest and mean temperature in May-June for gradient boosting. Mean temperature in March is also among the least influential variables for gradient boosting.

## Influence of temperature and precipitation on the probability of high disease severity

The gradient boosting algorithm is selected here because, in most cases, this type of model was the most accurate among the tested models including climate inputs (Fig 4). This model is

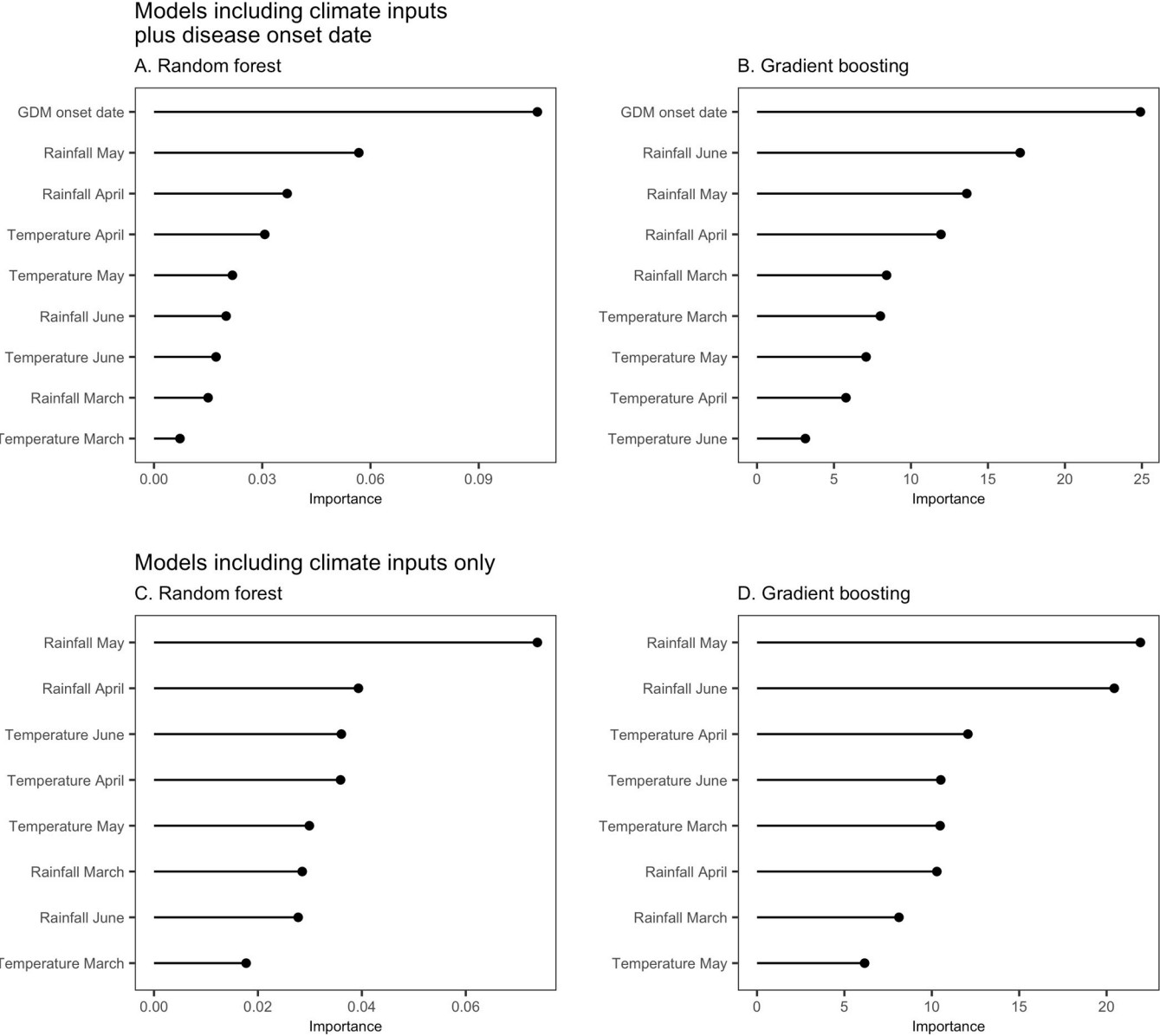

**Fig 5. Importance of the inputs used in random forest and gradient boosting models predicting the risk of high GDM severity on leaves at bunch closing stage.** Models presented in A and B include all inputs (date and climate) and models presented in C and D include climate inputs only. The importance metric reflects the gain in the model performance resulting from the use of each input.

thus used here to analyze the sensitivity of the probability of high disease severity on leaves to monthly temperature and precipitation changes.

The results are presented in Fig 6. Each graphic in Fig 6 shows the effect of precipitation (from -15% to +15%) and temperature (from +1˚C to 4˚C) change between the months of March and June. The effect of a fixed level of temperature increase during a given month, while keeping all other temperature variables unchanged is represented month-by-month in S4 Fig. The probability of high leaves severity shows an increasing trend as a function of precipitation increase (Fig 6). If we consider the precipitation change, the median probability of high severity increases from 0.57 to 0.79 for +15% of precipitation and +0% of temperature

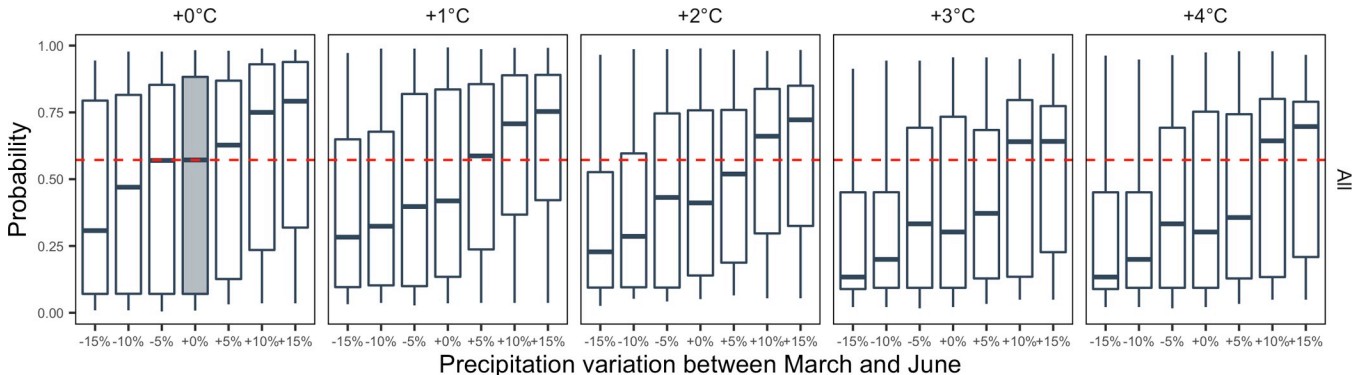

**Fig 6. Probability of high severity on leaves according to different precipitation variations and to different levels of temperature increase between March and June.** Each graphic shows the effect of precipitation change (from -15% to +15%) for a fixed level of temperature increase (from +0˚C to +4˚C) on predicted probability that GDM severity on leaves will be higher than regional median at the end of the season. Probabilities are forecasted by a gradient boosting algorithm that includes all climatic features. Each boxplot represents the distribution of the probability values over the vineyard plots of our dataset; the shaded boxplot corresponds to initial precipitation and temperatures (precipitation and temperature kept unchanged compared to actual conditions) and the median probability obtained with this scenario is indicated by a red dotted line. The lower and upper hinges of the boxes correspond to the first and third quartiles (the 25th and 75th percentiles) and vertical segment represent the range between min and max values.

(Fig 6). The first and third quartiles computed over the set of vineyard plots follow a similar increasing trend. Symmetrically, the probability of high severity decreases when the level of precipitation is reduced. Thus, a reduction of -15% of precipitation between March and June decreases the probability of high severity from 0.57 to 0.31. The strong sensitivity of the probability of high disease severity on leaves to precipitation is consistent with the increasing trend shown by the partial dependence plots displayed in Fig 7A and 7B for precipitation in May and June.

Overall, the temperature effect is smaller. Increasing temperature in April and May tends to have negative effect on the probability of high severity (S4 Fig). Thus, in May, the probability of high severity decreases from 0.57 (if the precipitation is kept unchanged) to 0.42 (at +3˚C). Even in case of a +15% increase of precipitation, the probability of high severity in May does not exceed its original value at +3˚C. The effect of temperature in June is positive but small (the probability increased from 0.57 to 0.76 at +4˚C). The sensitivity of the probability of high disease severity to temperature is consistent with the partial dependence plots obtained for

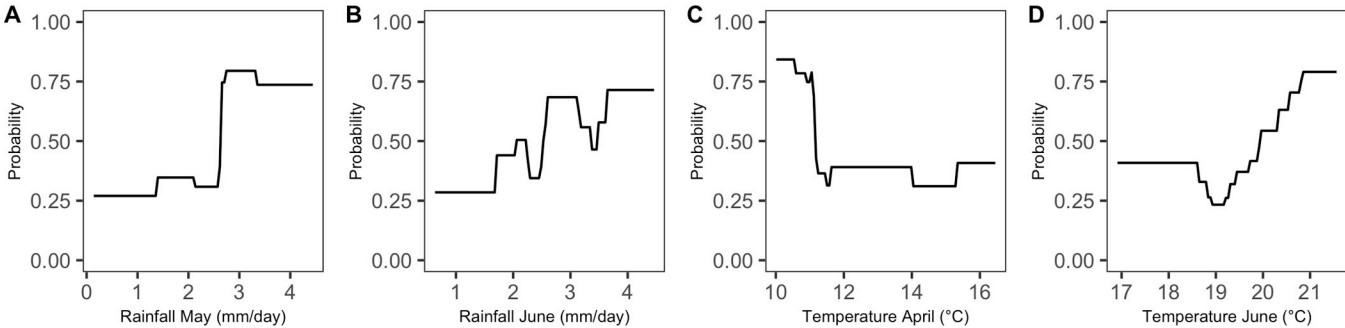

**Fig 7. Partial dependence plots of the relationships between probability of high GDM severity on leaves and the four most important climate variables of gradient boosting (according to Fig 5D).** Each graph represents the marginal effect of one variable on the probability computed by the gradient boosting model. (A) Partial dependence plot for the average amount of rainfall in May (in mm/day). (B) Partial dependence plot for the average amount of rainfall in June (in mm/day). (C) Partial dependence plot for the mean temperature in April (in ˚C). (D) Partial dependence plot for the mean temperature in June (in ˚C).

temperature (Fig 7C and 7D); these plots reveals a decreasing trend in April and an increasing trend in June but with a plateau covering a large range of temperatures.

## Potential reduction of GDM treatment

A late occurrence of GDM on vines resulted in a decrease of the probability of high incidence and severity. For example, the probability of high GDM severity on leaves computed by the GLM was higher than 0.75 in case of early disease onset (before week 22, i.e. late May, early June) but lower than 0.5 when disease onset occurred after week 24 (95%IC = [23.2; 25]), i.e. mid-June (Fig 8). The probability to reach high GDM incidence on leaves estimated by the GLM decreases from 0.92 (95%IC = [0.86; 0.99]) for a disease onset at week 19 (late May) to 0.5 (95%IC = [0.39; 0.61]) for a disease onset at week 24 (late June) (Fig 8). A similar decreasing trend was obtained with the other models, in particular with gradient boosting (Figs 8 and S5).

The probability of high disease severity and incidence computed as a function of disease onset date can be used to trigger fungicide treatments. We consider here a decision rule in which the first treatment is applied only (i) when disease symptoms are observed and (ii) when the probability of high disease incidence/severity estimated as a function of the date of disease onset exceeds a certain threshold. With this decision rule, when the threshold is zero, the first treatment is applied in a plot as soon as some disease symptoms are observed in that plot. If the threshold is set to a value higher than zero, only the plots exceeding the corresponding probability value will receive a treatment. The resulting average numbers of treatments are reported in Fig 9A. In this figure, the results are obtained with the probability of high disease severity on leaves, but very similar results are obtained with other response variables, i.e. probability of high disease incidence on leaves and on bunches, or the probability of high disease severity on bunches (see S6 Fig).

According to this rule, triggering the first spray as soon as the model output exceeds 0 leads to 5.1 treatments against the disease, in average. Fig 9A shows that the number of treatments decreases substantially when the considered probability threshold increased. For example, triggering first fungicide treatment when the probability of high GDM severity on leaves exceeds 0.5 leads to 3.7 treatments in average, which corresponds to a reduction of 1.4 application in average compared to a systematic application at disease onset. The setting of the probability threshold at 0.75 further reduces the average number of applications, i.e. to 1.5 treatments.

The corresponding percentage of treatment reduction are showed in Fig 9B. With probability thresholds of 0.5 and 0.75, the number of fungicide applications against GDM was lower by 27.2% and 70.2% compared to a systematic fungicide application at disease onset, respectively (Fig 9B).

This potential reduction of GDM treatments is even more important when compared to the current practices observed in Bordeaux vineyards. According to the results of a survey conducted by the SSP, 7.9 and 10.1 fungicide treatments against GDM were applied in average in Bordeaux vineyards in 2010 and 2013, respectively [61]. Compared to the average number of treatment values obtained in 2010 in the Bordeaux region, it is possible to reduce the number of treatments applied in 2010 by 53.3% and 80.9% by using our model-based decision rule with a probability threshold of 0.5 and 0.75, respectively. These levels of reduction reach 63.5% and 85.1% (with a probability threshold of 0.5 and 0.75, respectively) when considering the average number of treatments against GDM sprayed in 2013 in the Bordeaux region (Fig 9B).

## Discussion

In our analysis, we were able to relate the risk of high GDM incidence and severity on leaves and on bunches to the disease onset date and to climatic conditions in spring. Our study

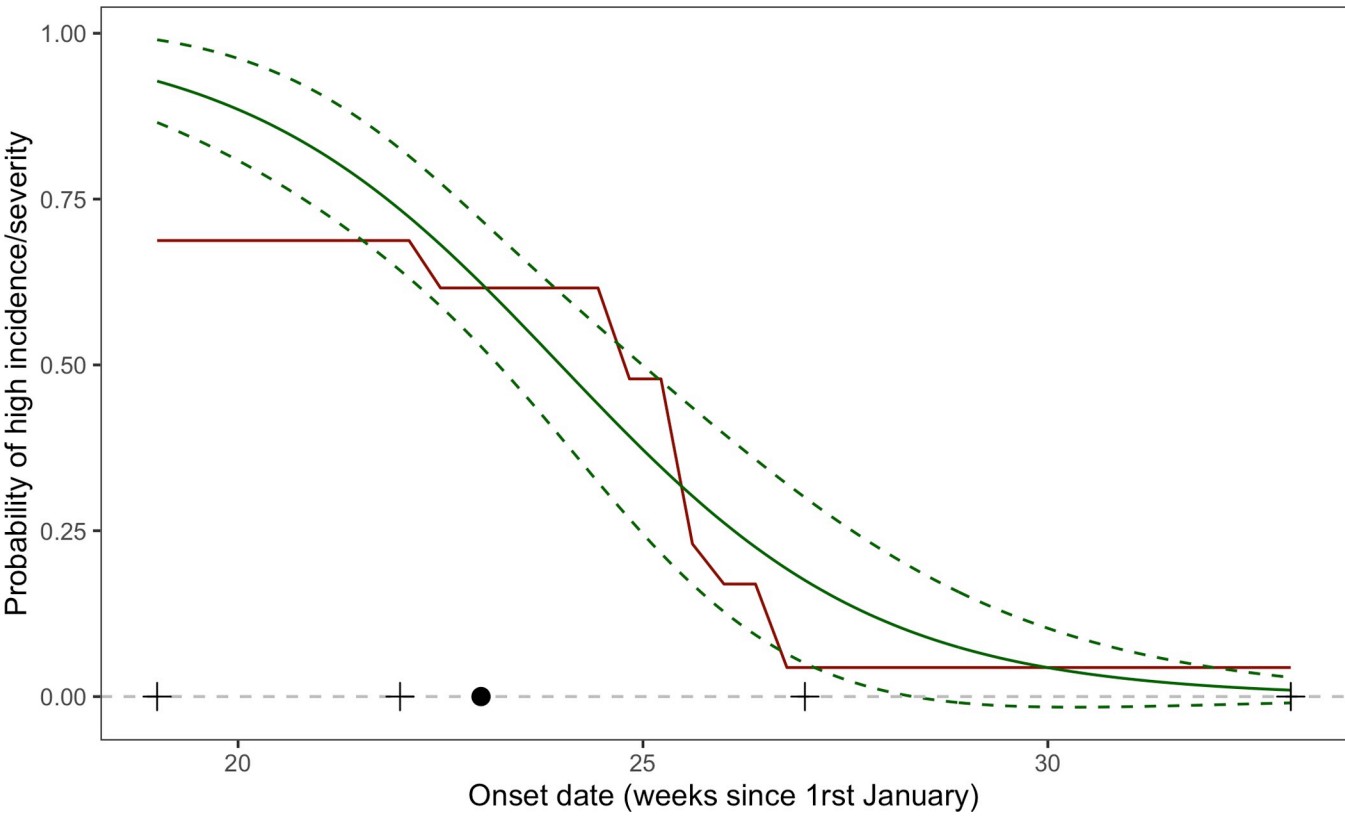

**Fig 8. Response of probability of high severity on leaves to date of disease onset estimated with the GLM and its 95% confidence interval (in green), and partial dependence plot obtained with the gradient boosting algorithm including climate inputs and date of disease onset (in red).** Median, minimum, 1[st] and 3[rd] quartiles, and maximum of observed onset dates are represented by a dot and four crosses, respectively.

shows that the date of appearance of GDM has a greater influence on GDM infection levels than climate variables. An early onset date, i.e. before the end of May or early June, leads to a higher probability of a strong attack, while later infections, i.e. after the end of June, are associated with a lower risk. This result is consistent with those of Kennelly et al. [62], who showed that late infection reduced the severity of GDM on bunches due to the development of ontogenic berry resistance. In addition, GDM is a polycyclic disease, which means that early first infection increases the number of asexual cycles and the infection rates.

The reason for the strong influence of the date of disease onset probably lies in the fact that this variable already integrates many factors, in particular climatic factors. Indeed, Chen et al. [47] showed that the date of disease onset depends on spring precipitation. This is consistent with the fact that, among the climatic factors, we found that spring precipitation was the most influential. More generally, the climate conditions at the end of spring, and more particularly in May, were found to be decisive for the development of GDM in the Bordeaux vineyards.

Our analysis showed that a decrease in spring precipitation leads to a reduction in the risk of GDM. The development of oospores, the main source of inoculum of primary infection, is inhibited by dry periods of spring [38]. Precipitation is also necessary for the dispersion and survival of GDM zoospores that cause infection in grape leaves, bunches and shoots [3]. The effect of temperature on GDM is more complex and depends on the period. Our results indicate that a temperature increase in late spring, i.e. June, tends to favor a high incidence or severity of GDM. On the contrary, an increase in temperature in April and May tends to

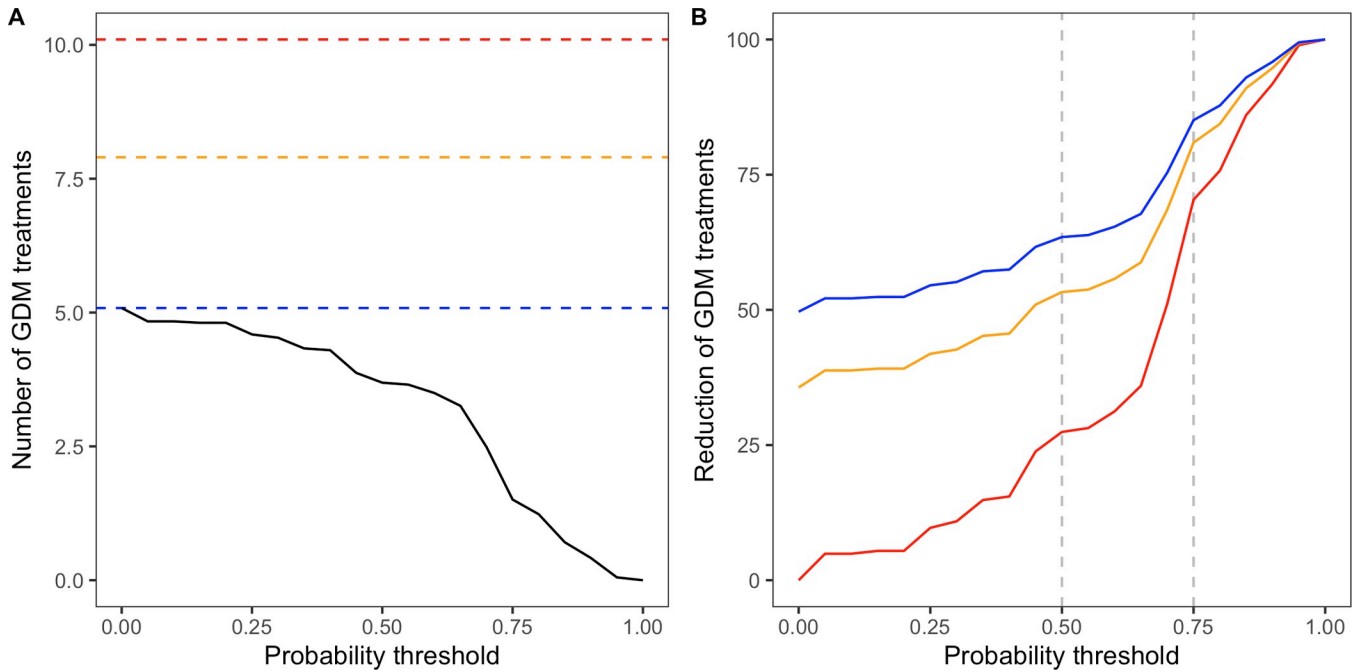

**Fig 9. Impact of the model-based decision rule on GDM pesticide use in Bordeaux vineyards as a function of a predefined triggering probability threshold (probability of high GDM severity on leaves).** (A) The black curve indicates the average numbers of fungicide treatments in the vineyard plots of our dataset computed while assuming that the first treatment is triggered only when the GLM probability of high severity exceeds the value given in the x-axis. Blue line represents the number of treatments for threshold = 0, i.e. when the first treatment is applied in all plots as soon as GDM symptoms are detected. Red and orange lines correspond to the average numbers of treatments recorded by the SSP in 2013 and 2010, respectively. (B) Potential reduction of GDM treatments compared to other treatment scenarios, represented by the color of each curve, and computed according to the predefined triggering probability threshold. Blue curve represents the reduction induced by the application of the decision rules compared to the strategy where first treatment is triggered at disease onset. Orange and red lines represent the treatment reduction compared to the results of SSP study in 2010 and 2013, respectively.

reduce the risk of a serious attack of GDM. The positive effect of an increased temperature in June on disease incidence and severity is consistent with the results of Salinari et al. [63] and Rossi et al. [64] who found that high temperature accelerates oospores germination.

The results of our sensitivity analysis suggest that a climate change scenario characterized by a decrease in precipitation and an increase in temperature in the spring reduces the risk of a serious attack on GDM. These results are consistent with those of Launay et al. [65], who show that a lower risk of infection with GDM can be expected in regions with oceanic climatic conditions, such as Bordeaux vineyards, in case of temperature increase and reduced leaf moisture duration. On the other hand, according to Salinari et al. [63], increased air temperature and reduced precipitation in Bordeaux vineyards would advance the first symptoms of GDM, which could lead to a more serious infection due to the polycyclic nature of the pathogen. It should be noted that Salinari et al. [63] considered a climate scenario in which the decrease in precipitation was insufficient to reduce the risk of GDM. Potentially, our models could thus be used to adapt grape disease management in different contexts of climatic change [66]. However, several factors are not taken into account by our statistical and machine learning tools. The high potential of adaptability of plants pathogen under new climatic conditions is not considered in our study [67]. Climatic inputs were limited to aggregated precipitation and temperature conditions during spring only. Other climatic variables such as solar radiation, moisture, hydro-thermal time, which are used in other epidemiological models [31,40], are omitted in our models. However, our approach is simple to implement as soon as disease observations are available. It could be applied to other pathogens for which such data are collected [68] like

septoria leaf blotch of wheat [69]. Our models can be easily updated with new observational data, providing an additional level of confidence to end users in terms of model accuracy [70].

Our models could also be used for another type of application; they could be integrated in decision support systems to reduce the number of fungicide treatments in Bordeaux vineyards, where more than 10 treatments are commonly sprayed to control GDM [11]. Forecasts of our models could be used to trigger treatment when the predicted risk of GDM at bunch closure is higher than a certain threshold, in order to avoid unnecessary fungicide treatments. Based on our model assessment, the most accurate model is the one including all features, i.e., both date of disease onset and climate inputs. However, this model poses some practical problems. As some of its climate inputs are not available before June, this model could be used in late spring only, i.e., after the start of the GDM epidemic. Since the omission of the climate variables from the set of inputs only slightly reduces the performance of the models, we recommend using the version of the GLM model including the date of disease onset as the only input variable. This model can be used as soon as the presence of GDM is detected and does not require any climate variables. A drawback of this approach is that it involves constant and frequent field scouting in order to determine GDM onset date. In the future, sensors on drones may become available for automatic disease detection [71,72].

In Bordeaux vineyards, we show that more than 50% of the treatments against GDM could be avoided compared to current practices if GLM forecast were used to trigger first fungicide application. This result is based on the decision rule established in our study from GLM predictions. Following this rule, the first treatment is triggered if the predicted probability of a severe attack is higher than a given threshold. In average, the application of this decision rule (with a probability threshold equal to 0.5) resulted in a 53% and 63% reduction of the number of treatments against the disease, compared to the average number of treatments reported in the surveys conducted in the Bordeaux regions in 2010 and 2013, respectively. Our results are consistent with several previous studies conducted in other major vine producing countries. Several warning systems were indeed developed to identify periods when conditions are favorable for GDM development (infection or sporulation), and to schedule necessary fungicide applications [21,73,74]. It was shown that the implementation of these tools could lead to a reduction in the number of fungicide applications compared to current practices. For example, the warning system developed by Caffi et al. [21] led to a median reduction of 54% of the number of fungicide applications, compared to standard schedules in Italian vineyards. Similar results were obtained by Pellegrini et al. [74] and Menesatti et al. [40].

The practicality of our approach should be assessed in close collaboration with farmers and agricultural extension services. The proposed approach requires field scouting, which is time and labor consuming for winegrowers, but in the future, observation costs could be strongly decreased by using automatic disease detection methods [75], such as image analysis [71] or airborne inoculum detection [76]. These recent techniques are likely to reduce the cost of symptom detection. Furthermore, although delaying fungicide treatments can be perceived as risky by some grape growers, very few experiments have been conducted to support this statement. The study by Menesatti et al. [40] challenged this perception and showed that a strategy based on triggering fungicide application at GDM onset contributed to effectively control the disease and to reduce the number of fungicide applications by almost half, compared to current control practices in Italian organic vineyards. However, as the number of available experimental studies is limited, new experiments covering a variety of agricultural and environmental conditions would be useful to assess more precisely the potential economic benefits and risks of this strategy. Crop insurance could also be offered to producers as a mean of covering the GDM risk associated with the use of decision support tools delaying the first fungicide treatments [77]. Our approach could also be assimilated as an insurance index to

offer a market-based method of reducing the overuse or inefficient use of fungicides [77]. Although the systematic use of fungicide treatments currently appears to be an effective solution for controlling GDM, regulations on pesticide use may become more restrictive in the future, forcing grape growers to reduce their use of fungicides.

## Supporting information

**S1 Data. Models inputs.**
(XLSX)

**S1 Fig.** Grape downy mildew (GDM) incidence data on leaves after bunch closing (A) and imputed disease onset dates (B) in 151 untreated plots. Median contamination levels and median disease onset date are represented by vertical dotted lines.
(PNG)

**S2 Fig.** Grape downy mildew (GDM) incidence data on bunches after bunch closing (A) and imputed disease onset dates (B) in 156 untreated plots. Median contamination levels and median disease onset date are represented by vertical dotted lines.
(PNG)

**S3 Fig.** Grape downy mildew (GDM) severity data on bunches after bunch closing (A) and imputed disease onset dates (B) in 152 untreated plots. Median contamination levels and median disease onset date are represented by vertical dotted lines.
(PNG)

**S4 Fig. Probability of high severity on leaves according to different precipitation variations (lines) and to different levels of temperature increase between March and June (columns).** Each graphic shows the effect of precipitation change during a given period (from -15% to +15% between March and June, in April, in May, or in June) for a fixed level of temperature increase (from +0°C to +4°C between March and June) on predicted probability that GDM severity on leaves will be higher than regional median at the end of the season. Probabilities are forecasted by a gradient boosting algorithm that includes all climatic features. Each boxplot represents the distribution of the probability values over the vineyard plots of our dataset; the shaded boxplot corresponds to initial precipitation and temperatures (precipitation and temperature kept unchanged compared to actual conditions) and the median probability obtained with this scenario is indicated by a red dotted line. The lower and upper hinges of the boxes correspond to the first and third quartiles (the 25th and 75th percentiles) and vertical segment represent the range between min and max values.
(PNG)

**S5 Fig. Response of probability of high incidence or severity on leaves or on bunches to date of disease onset estimated with the GLM and its 95% confidence interval (in green), and partial dependence plot obtained with the gradient boosting algorithm including climate inputs and date of disease onset (in red).** Median, minimum, 1st and 3rd quartiles, and maximum of observed onset dates are represented by a dot and four crosses, respectively.
(PNG)

**S6 Fig. Number of fungicide treatments applied to control GDM in Bordeaux vineyards as a function of a predefined triggering probability threshold (probability of high GDM incidence or severity on leaves or on bunches).** The black curve indicates the average numbers of fungicide treatments in the vineyard plots of our dataset computed while assuming that the first treatment is triggered only when the GLM probability of high severity exceeds the value

given in the x-axis. Blue line represents the number of treatments for threshold = 0, i.e. when the first treatment is applied in all plots as soon as GDM symptoms are detected. Red and orange lines correspond to the average numbers of treatments recorded by the SSP in 2013 and 2010, respectively.
(PNG)

## Acknowledgments

We thank the French Vine and Wine Institute (*Institut Français de la Vigne et du Vin*) and its technical partners for collecting and providing us with access to their data and to the EPIcure web platform (https://www.vignevin-epicure.com). We also thank M. Vergnes for data collection coordination and C. Debord for database administration.

## Author Contributions

**Conceptualization:** Mathilde Chen, François Brun, Marc Raynal, David Makowski.

**Formal analysis:** Mathilde Chen, David Makowski.

**Validation:** François Brun, Marc Raynal, David Makowski.

**Writing – original draft:** Mathilde Chen, David Makowski.

**Writing – review & editing:** Mathilde Chen, David Makowski.

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
