## [Decision Letter · Decision Letter 0]

12 Dec 2019

PONE-D-19-22557

Forecasting severe grape downy mildew attacks using machine learning

PLOS ONE

Dear Dr. Mathilde Chen,

Thank you for submitting your manuscript to PLOS ONE. After careful consideration, we feel that it has merit but does not fully meet PLOS ONE’s publication criteria as it currently stands. Therefore, we invite you to submit a revised version of the manuscript that addresses the points raised during the review process.

Both Reviewers indicate the need to improve the paper significantly.

We would appreciate receiving your revised manuscript by Jan 26 2020 11:59PM. To enhance the reproducibility of your results, we recommend that if applicable you deposit your laboratory protocols in protocols.io, where a protocol can be assigned its own identifier (DOI) such that it can be cited independently in the future. For instructions see: http://journals.plos.org/plosone/s/submission-guidelines#loc-laboratory-protocols

We look forward to receiving your revised manuscript.

Kind regards,

Andrea Luvisi

Academic Editor

PLOS ONE

Journal Requirements:

Additional Editor Comments (if provided):

Reviewers' comments:

Reviewer's Responses to Questions

**Comments to the Author**

1. Is the manuscript technically sound, and do the data support the conclusions?

Reviewer #1: Partly

Reviewer #2: Yes

2. Has the statistical analysis been performed appropriately and rigorously? 

Reviewer #1: Yes

Reviewer #2: No

3. Have the authors made all data underlying the findings in their manuscript fully available?

Reviewer #1: Yes

Reviewer #2: Yes

4. Is the manuscript presented in an intelligible fashion and written in standard English?

Reviewer #1: No

Reviewer #2: Yes

5. Review Comments to the Author

Reviewer #1: This manuscript explores a machine-learning model for predicting Grape Downy Mildew incidence and severity with the hope that growers can use this information to reduce the number of fungicide applications. After testing several models and algorithms, the authors find that date of disease onset is the best predictor of GDM incidence and severity, along with some climatic factors. The authors present an in-depth look at constructing a model, but the practicality of the model they have produced is lost. They acknowledge that growers will most likely not adopt practices involving increased scouting or delayed spraying.

Spelling and grammar need to be corrected throughout.

113 and 114: Define site-year

115: 3 plants is very few. Is this data reliable? That is a very large range in plot size.

121: 1 to 57 is a very large range in number of site visits. Is this data reliable?

128: What do you mean by vine stocks?

147: Mislabeled as Fig. 1

226: Mislabeled as Fig. 2

272: Use of the term random forest twice

272: Mislabeled as Fig. 3

286: Mislabeled as Fig. 4

377: Use of attack is unusual – perhaps infection?

441: How this would reduce treatments by more than 50% is still unclear. This is a bold claim and I feel that it requires more detail so that readers can understand how the authors reached this number. It would be helpful to see an additional paragraph on how this number was derived.

Fig 1. Caption does not mention incidence and does not offer enough detail on analysis. Medium severity looks very close to 0 – figures should be formatted to better show the severity instead of using the same scale as incidence. There is no mention of severity in the text, only incidence (Line 126) – describe the results in the text.

Fig 2. Put labels in English. Lines are not distinguishable. What does this figure contribute to the manuscript?

Fig. 6. Provide details of statistical analysis. More detail required.

Fig. 7. This caption requires significantly more detail.

Fig. 9. This caption and figure are unclear to me yet are the basis for the claim that fungicide applications can be reduced by 50%. More detail and explanation is required so that we understand this claim fully, because it has big implications.

Reviewer #2: The authors study the ability of machine learning algorithms to predict the onset of GDM to reduce the number of fungicide treatments. The analysis is performed on an extensive data set spanning many years from the Bordeaux region. The problem is well motivated. The English is very good with only a few very minor mistakes. But, I have some open questions about the technical/ML approach. I believe the paper is at least a major revision.

Major:

Novelty is an important part of the publication process in general. Can the authors specifically comment on the novelty of this work? This is especially important because there are some cited works (36-39) that seem similar. It should not be up to the readers to infer the novelty of the work.

How is the GDM onset date encoded as an input/feature? It would be a date, which is a character string, it’s not obvious how a study would numerical-ize this for a machine learning problem, and the authors should state this.

I’m confused by the choice of ROC-AUC as a metric for measuring performance. The problem is a regression problem, and AUC is typically used for classification problems. Traditionally variations of squared error are used with regression problems. Is there a literature basis for using AUC for a regression problem? Currently, the set up for the experiment is this:

1) Regression analysis

2) Cast the problem as a classification problem by thresholding based on the median

3) Get AUC/ROC

Why do regression in the first place if this was the end goal?

I’m not sure I agree on thresholding the output based on greater/less than the median. This would only detect if a system is over/underpredicting GDM in a magnitude-less way. Isn’t the goal to get an accurate probability? You cannot measure error well if you do this.

Can the authors elaborate one Line 186 what they mean by ‘median’ here? Is this the median value of the plot as a whole, median of the year, etc.?

The outputs (incidence and severity) seem subjective because an expert would have to eyeball the spread of disease. I’m surprise the authors continued with some plots that had only 1 data inspection.

The abstract states the authors use a year-by-year cross validation, this isn't a conventional cross-validation method and needs to be explained. Yet, it is not in the methods section.

Very minor:

Methods section: It’s conventional to state the number of input values for your data set, the readers should not infer this from a later figure.

Line 53: I think this should be “Leaf” and not “Leaves”

Line 83: Is mechanistic models the right term for this? Probabalistic models still need to determine some function of a model.

Fig. 1: Box plots are helpful, but the authors should state the boundaries of the box. Is this the percentile? If so please state what they are.

Figure 2: Axes are in French.

Line 175: Technically, 100 trees is a parameter of the algorithm, this statement is not true in general.

Line 232: Sever attack -> severe attack

6. PLOS authors have the option to publish the peer review history of their article (what does this mean?). If published, this will include your full peer review and any attached files.

Reviewer #1: No

Reviewer #2: Yes: Alberto C. Cruz, Ph.D.

---

## [Author Response · Author response to Decision Letter 0]

1 Feb 2020

Reviewer #1: 

This manuscript explores a machine-learning model for predicting Grape Downy Mildew incidence and severity with the hope that growers can use this information to reduce the number of fungicide applications. After testing several models and algorithms, the authors find that date of disease onset is the best predictor of GDM incidence and severity, along with some climatic factors. The authors present an in-depth look at constructing a model, but the practicality of the model they have produced is lost. They acknowledge that growers will most likely not adopt practices involving increased scouting or delayed spraying.

Spelling and grammar need to be corrected throughout.

Answer: The proposed approach requires field scouting, which is time and labor consuming for winegrowers, but in the future, observation costs could be strongly decreased by using automatic disease detection methods (Martinelli et al., 2015), such as image analysis (Rieder et al., 2014) or airborne inoculum detection (Thiessen et al., 2016). These recent techniques are likely to reduce the cost of symptom detection. Furthermore, although delaying fungicide treatments can be perceived as risky by some grape growers, very few experiments have been conducted to support this statement. The study by Menesatti et al. (2015) challenged this perception and showed that a strategy based on triggering fungicide application at GDM onset contributed to effectively control the disease and to reduce the number of fungicide applications by almost half, compared to current control practices in Italian organic vineyards. However, as the number of available experimental studies is limited, new experiments covering a variety of agricultural and environmental conditions would be useful to assess more precisely the potential economic benefits and risks of this strategy. Crop insurance could also be offered to producers as a mean of covering the GDM risk associated with the use of decision support tools delaying the first fungicide treatments (Norton et al., 2016). Our approach could also be assimilated as an insurance index to offer a market-based method of reducing the overuse or inefficient use of fungicides (Norton et al., 2016). 

This is now specified in the discussion section of the revised manuscript. See lines 540 to 557. 

113 and 114: Define site-year

Answer: Different vineyards were included in the dataset each year. A site-year is a unique combination of vineyard site and year. This is now specified in the revised manuscript. See lines 124 to 125.

115: 3 plants is very few. Is this data reliable? That is a very large range in plot size.

Answer: Plots including less than 6 plants (only two plots) were excluded from the dataset. It is now mentioned line 127. The new dataset was used in the revised paper and the results were updated. New results are almost identical compared to the original results.

121: 1 to 57 is a very large range in number of site visits. Is this data reliable?

Answer: These values were updated. In total, 1 to 19 visual inspections were conducted in each vineyard. See line 133. 

128: What do you mean by vine stocks?

Answer: In this study, dates of GDM onset were estimated by analyzing incidence data on vine plants, i.e. the proportion of infected plants in a plot. This is now clarified in the revised manuscript. See lines 144 to 145. 

147: Mislabeled as Fig. 1

Answer: Fig 1 caption was rewritten in accordance with PLOS ONE authors’ guideline concerning figures’ caption. See lines 157 to 163. 

226: Mislabeled as Fig. 2

Answer: Fig 2 caption was rewritten in accordance with PLOS ONE authors’ guideline concerning figures’ caption. See lines 175 to 180. 

272: Use of the term random forest twice

Answer: The phrase was corrected. See line 335. 

272: Mislabeled as Fig. 3

Answer: Fig 3 caption was rewritten in accordance with PLOS ONE authors’ guideline concerning figures’ caption. See lines 269 to 276. 

286: Mislabeled as Fig. 4

Answer: Fig 4 caption was rewritten in accordance with PLOS ONE authors’ guideline concerning figures’ caption. See lines 323 to 324. 

377: Use of attack is unusual – perhaps infection?

Answer: The phrase was rewritten. See line 458. 

441: How this would reduce treatments by more than 50% is still unclear. This is a bold claim and I feel that it requires more detail so that readers can understand how the authors reached this number. It would be helpful to see an additional paragraph on how this number was derived.

Answer: In Bordeaux vineyards, we show that more than 50% of the treatments against GDM could be avoided compared to current practices if GLM forecast were used to trigger first fungicide application. This result is based on the decision rule established in our study from GLM predictions. Following this rule, the first treatment is triggered if the predicted probability of a severe attack is higher than a given threshold. In average, the application of this decision rule (with a probability threshold equal to 0.5) resulted in a 53% and 63% reduction of the number of treatments against the disease, compared to the average number of treatments reported in the surveys conducted in the Bordeaux regions in 2010 and 2013, respectively. Our results are consistent with several previous studies conducted in other major vine producing countries. This is now detailed in the discussion section of the revised manuscript. See lines 521 to 530. 

Besides, more details on these results were added to the Result section of the revised paper. See lines 432 to 441 and the new Fig 9. 

Fig 1. Caption does not mention incidence and does not offer enough detail on analysis. Medium severity looks very close to 0 – figures should be formatted to better show the severity instead of using the same scale as incidence. There is no mention of severity in the text, only incidence (Line 126) – describe the results in the text.

Answer: Fig 1 was improved. See new Fig 1, its caption lines 157 to 163. Information on median GDM severity was added. See lines 140 to 141.

Fig 2. Put labels in English. Lines are not distinguishable. What does this figure contribute to the manuscript?

Answer: Axis labels were translated in English. Fig 2 was revised to give more information on annual variability in the climatic dataset. See new Fig 2. 

Fig. 6. Provide details of statistical analysis. More detail required.

Answer: Results presented in the original Fig6 were split between the new Fig6 and new S5 figure. Each graphic in new Fig 6 shows the effect of precipitation (from -15% to +15%) and temperature (from +1°C to 4°C) change between the months of March and June. The effect of a fixed level of temperature increase during a given month, while keeping all other temperature variables unchanged is represented month-by-month in S5 figure. Details were added to the caption of new Fig 6. See lines 363 to 372. 

Fig. 7. This caption requires significantly more detail.

Answer: Details were added to the caption of Figure 7. See lines 384 to 389. 

Fig. 9. This caption and figure are unclear to me yet are the basis for the claim that fungicide applications can be reduced by 50%. More detail and explanation is required so that we understand this claim fully, because it has big implications.

Answer: We consider here a decision rule in which the first treatment is applied only (i) when disease symptoms are observed and (ii) when the probability of high disease incidence/severity estimated as a function of the date of disease onset exceeds a certain threshold. The number of treatments depend on the selected probability threshold.

New Fig 9A shows that the number of treatments decreases substantially when the considered probability threshold increased. For example, triggering first fungicide treatment when the probability of high GDM severity on leaves exceeds 0.5 leads to 3.7 treatments in average, which corresponds to a reduction of 1.4 application in average compared to a systematic application at disease onset. The setting of the probability threshold at 0.75 further reduces the average number of applications, i.e. to 1.5 treatments. 

The corresponding percentage of treatment reduction are showed in Fig 9B. With probability thresholds of 0.5 and 0.75, the number of fungicide applications against GDM was lower by 27.2% and 70.2% compared to a systematic fungicide application at disease onset, respectively (new Fig 9B). 

This potential reduction of GDM treatments is even more important when compared to the current practices observed in Bordeaux vineyards. According to the results of a survey conducted by the SSP, 7.9 and 10.1 fungicide treatments against GDM were applied in average in Bordeaux vineyards in 2010 and 2013, respectively (Service de la Statistique et de la Prospection, 2016). Compared to the average number of treatment values obtained in 2010 in the Bordeaux region, it is possible to reduce the number of treatments applied in 2010 by 53.3% and 80.9% by using our model-based decision rule with a probability threshold of 0.5 and 0.75, respectively. These levels of reduction reach 63.5% and 85.1% (with a probability threshold of 0.5 and 0.75, respectively) when considering the average number of treatments against GDM sprayed in 2013 in the Bordeaux region (new Fig 9B). 

This is now detailed in the results section of the revised manuscript. See lines 420 to 441. 

Fig 9 and its caption were improved in order to give more details on how we obtained this value. See new Fig 9 and lines 442 to 452. 

 

Reviewer #2: 

The authors study the ability of machine learning algorithms to predict the onset of GDM to reduce the number of fungicide treatments. The analysis is performed on an extensive data set spanning many years from the Bordeaux region. The problem is well motivated. The English is very good with only a few very minor mistakes. But, I have some open questions about the technical/ML approach. I believe the paper is at least a major revision.

Major:

Novelty is an important part of the publication process in general. Can the authors specifically comment on the novelty of this work? This is especially important because there are some cited works (36-39) that seem similar. It should not be up to the readers to infer the novelty of the work.

Answer: In this study, we assess the ability of statistical models and machine learning algorithms to predict the occurrence of high GDM levels at the end of the season, which has never been done before. More specifically, we develop different statistical and machine learning models to predict the risk of high GDM incidence or severity on leaves and bunches at the end of the season, in untreated Bordeaux vineyards. The models tested are generalized linear models (Nelder et al., 1972), regularized regression models (LASSO) (Tibshirani, 1996), and two machine learning algorithms, i.e. gradient boosting (Friedman, 2002) or random forests (Breiman, 2001). This is now clearly mentioned in the revised version. See lines 103 to 109. 

How is the GDM onset date encoded as an input/feature? It would be a date, which is a character string, it’s not obvious how a study would numericalize this for a machine learning problem, and the authors should state this.

Answer: In this study, dates of GDM onset were estimated by analyzing incidence data on vine plants, i.e. the proportion of infected plants in a plot. GDM onset was defined as the first week in which the proportion of infected vines stocks exceeded 1%. The number of weeks between the first week of the year and this date was estimated for each plot by survival analysis in order to deal with censored data (Chen et al., 2018). This is now specified in the revised manuscript. See lines 144 to 148.

I’m confused by the choice of ROC-AUC as a metric for measuring performance. The problem is a regression problem, and AUC is typically used for classification problems. Traditionally variations of squared error are used with regression problems. Is there a literature basis for using AUC for a regression problem? Currently, the set up for the experiment is this:

1) Regression analysis (nous on fait une regression binomial adaptée à la classification binaire → pas de regression)

2) Cast the problem as a classification problem by thresholding based on the median

3) Get AUC/ROC

Why do regression in the first place if this was the end goal?

Answer: In this study, we don’t perform any quantitative regression. We use binomial regression models (binomial-logistic GLM and binomial-LASSO) and classification methods (gradient boosting, random forest) to calculate the probability to reach a high level of contamination at bunch closing stage, i.e. higher than the median value reported in the dataset. The target variable is binary (1 for high disease level, 0 for low disease level) and indicates whether a high disease level was reached or not in each plot. 

The ability of the fitted models to predict occurrence of high level of GDM was assessed by the area under the ROC curve (AUC). This criterion is commonly used for comparing the performances of binomial regression models and classification methods based on machine learning. See for examples: 

- Fawcett, T., 2006. An introduction to ROC analysis. Pattern Recognition Letters, 27(8):861–874.

- Bradley, A. P., 1997. The use of the area under the ROC curve in the evaluation of machine learning algorithms. Pattern Recognition, 30, 1145–1159

I’m not sure I agree on thresholding the output based on greater/less than the median. This would only detect if a system is over/underpredicting GDM in a magnitude-less way. Isn’t the goal to get an accurate probability? You cannot measure error well if you do this.

Answer: In this study (as in many plant health studies), we do not attempt to quantitatively predict GDM incidence or severity values. Our objective is to estimate whether GDM attack will exceed a given threshold of incidence or severity at the end of the season. In this study, the considered threshold for each output is the regional median, computed from the 153 plots included in the dataset. However, our method is generic and other thresholds could be considered. This approach is common in phytosanitary studies when the objective is to discriminate between low and high levels of infection. For examples, see:

- Yuen J, Twengstrom E, Sigvald R, 1996. Calibration and verification of risk algorithms using logistic regression. European Journal of Plant Pathology 102, 847–54.

- Hughes, G., McRoberts, N., and Burnett, F.J., 1999. Decision-making and diagnosis in diseases management. Plant Pathology 48:147-153.

- Makowski, D., Taverne M., Bolomier J., Ducarne M., 2005. Comparison of risk indicators for sclerotinia control in oilseed rape. Crop Protection 24:527-531.

These references were included in the revised paper. See line 247.

Can the authors elaborate one Line 186 what they mean by ‘median’ here? Is this the median value of the plot as a whole, median of the year, etc.?

Answer: In this study, the term "median" refers to the regional median over all years included in the dataset. These medians are computed from the 153 monitored plots. This is now clarified in the revised manuscript. See lines 227 to 230. 

The outputs (incidence and severity) seem subjective because an expert would have to eyeball the spread of disease. I’m surprise the authors continued with some plots that had only 1 data inspection.

Answer: In this study, we considered the impact of GDM at the end of the season, i.e. after bunch closing. For each plot, the level of GDM (i) leaf incidence, (ii) bunch incidence, (iii) leaf severity and (iv) bunch severity at the end of the season were derived from the last epidemiologic observations. 

In plots where one single observation was recorded, the inspection was carried out after bunch closing stage and this observation was thus relevant to estimate GDM level at the end of the season. In such plot, GDM onset date is, of course, censored. GDM onset date is also censored in plots including several observations when these observations are not collected every week. The number of censored data in our dataset is thus relatively large (61.7%). However, this issue was already discussed in a previous study and it was shown that censored date of GDM onset could be imputed by survival analysis (Chen et al., 2018) 

Here, censored date of disease onset were thus imputed using a semi-parametric survival model (Anderson-Bergman, 2017) including the average rainfall between March and June as covariate (Chen et al., 2018). In plots where few observations were collected, the imputed dates of disease onset were close to the median onset date of the dataset, which corresponds to week 23, i.e. early-mid June. See lines 148 to 151 in the material and methods section of the revised manuscript. 

The abstract states the authors use a year-by-year cross validation, this isn't a conventional cross-validation method and needs to be explained. Yet, it is not in the methods section.

Answer: The ability of the fitted models to predict occurrence of high level of GDM was assessed by year-by-year cross validation using the area under the ROC curve as a measure of classification performance (Barbottin et al., 2008; Makowski et al., 2009). Here, we used a year-by-year cross-validation to account for the strong "year" effect on disease intensity. As data collected the same year in different plots are not independent, it was safer to remove all the data collected a given year at each cross-validation step. This is equivalent to a group-wise cross-validation based on 9 groups corresponding to the 9 years of data included in our dataset. This is now specified in the material and methods section of the revised manuscript. See lines 219 to 226.

Very minor:

Methods section: It’s conventional to state the number of input values for your data set, the readers should not infer this from a later figure.

Answer: OK. See lines 194 to 198. 

Line 53: I think this should be “Leaf” and not “Leaves”

Answer: OK. See line 53. 

Line 83: Is mechanistic models the right term for this? Probabalistic models still need to determine some function of a model.

Answer: Mechanistic models differ from traditional statistical models by the need to translate every stage of the development cycle of an organism as functions; their structure makes explicit hypotheses about the biological mechanisms that drive infection dynamics (Lessler et al., 2016). This type of model relies on the estimation of many parameters and requires a good knowledge of the biological mechanisms and of the impact of different environmental variables on these mechanisms. See lines 83 to 88.

Fig. 1: Box plots are helpful, but the authors should state the boundaries of the box. Is this the percentile? If so please state what they are.

Answer: The lower and upper hinges of the boxes correspond to the first and third quartiles (the 25th and 75th percentiles) and horizontal segment represent the range between min and max values. See the new caption of Fig 1. See lines 157 to 163. 

Figure 2: Axes are in French.

Answer: The text was translated in English. See new Fig 2. 

Line 175: Technically, 100 trees is a parameter of the algorithm, this statement is not true in general.

Answer: This paragraph was modified. See line 211. 

Line 232: Sever attack -> severe attack

Answer: Ok. See line 284.

---

## [Decision Letter · Decision Letter 1]

26 Feb 2020

Forecasting severe grape downy mildew attacks using machine learning

PONE-D-19-22557R1

Dear Dr. Chen

We are pleased to inform you that your manuscript has been judged scientifically suitable for publication and will be formally accepted for publication once it complies with all outstanding technical requirements.

With kind regards,

Andrea Luvisi

Academic Editor

PLOS ONE

Additional Editor Comments (optional):

Reviewers' comments:

Reviewer's Responses to Questions

**Comments to the Author**

1. If the authors have adequately addressed your comments raised in a previous round of review and you feel that this manuscript is now acceptable for publication, you may indicate that here to bypass the “Comments to the Author” section, enter your conflict of interest statement in the “Confidential to Editor” section, and submit your "Accept" recommendation.

Reviewer #1: All comments have been addressed

Reviewer #2: All comments have been addressed

2. Is the manuscript technically sound, and do the data support the conclusions?

Reviewer #1: Yes

Reviewer #2: Yes

3. Has the statistical analysis been performed appropriately and rigorously? 

Reviewer #1: Yes

Reviewer #2: Yes

4. Have the authors made all data underlying the findings in their manuscript fully available?

Reviewer #1: Yes

Reviewer #2: Yes

5. Is the manuscript presented in an intelligible fashion and written in standard English?

Reviewer #1: Yes

Reviewer #2: Yes

6. Review Comments to the Author

Reviewer #1: Line 130-131 and in Figure 3: The revised manuscript still uses the term ‘grape stock’, which is not a commonly used term in viticulture. Are the authors = referring to trunks? Canes? Shoots?

Reviewer #2: The manuscript now clearly states the novelty, and it is an impactful study on the ability to advise growers on fungicide treatments. The greatest strength of the manuscript is it’s discussion—rarely do statistical/machine learning works go into depth about “why” the model works. I have a few minor points, I believe the manuscript should be accepted (and no more than a minor revision).

It’s clear now that the problem is a binary classification task. The AUC is indeed appropriate for this, but a statistical/machine learning problem *must* provide more than just the AUC. With R, it should be trivial to provide the additional metrics: true positive rate, false positive rate, positive predictive value, F1 score, confusion matrix, etc. (average over year-folds?). In particular, positive predictive value will indicate the promise of this work as a diagnostic tool, whereas AUC is more of a measurement of classification performance. Is it possible to also provide the ROC graphs for each method as well?

Can the authors explicitly provide the a-priori rates for the classification task (preferably year-by-year)?

Line 149: How did the authors chose a 1% infection rate? Is this based on some prior work, or is it a parameter?

7. PLOS authors have the option to publish the peer review history of their article (what does this mean?). If published, this will include your full peer review and any attached files.

Reviewer #1: No

Reviewer #2: Yes: Alberto C Cruz

---

## [Editor Report · Acceptance letter]

2 Mar 2020

PONE-D-19-22557R1 

Forecasting severe grape downy mildew attacks using machine learning 

Dear Dr. Chen:

I am pleased to inform you that your manuscript has been deemed suitable for publication in PLOS ONE. Congratulations! Your manuscript is now with our production department. 

With kind regards,

on behalf of

Dr. Andrea Luvisi 

Academic Editor

PLOS ONE